# Analyzing Strawberry Preferences: Best–Worst Scaling Methodology and Purchase Styles

**DOI:** 10.3390/foods13101474

**Published:** 2024-05-10

**Authors:** Antonina Sparacino, Selene Ollani, Lorenzo Baima, Michael Oliviero, Danielle Borra, Mingze Rui, Giulia Mastromonaco

**Affiliations:** Department of Agricultural, Forest, and Food Sciences, University of Turin, Largo Paolo Braccini 2, Grugliasco, 10095 Turin, Italy; selene.ollani@unito.it (S.O.); lorenzo.baima@unito.it (L.B.); michael.oliviero@edu.unito.it (M.O.); danielle.borra@unito.it (D.B.); mingze.rui@unito.it (M.R.); giulia.mastromonaco@unito.it (G.M.)

**Keywords:** consumers’ choices, soft fruit, choice experiment, fruit aesthetics, health benefits, taste

## Abstract

This research has investigated Italian consumers’ preferences for and purchasing behaviors of strawberries utilizing the Best–Worst Scaling methodology (BWS). This approach enables the key factors that influence strawberry purchasing decisions to be identified and different choice groups to be characterized. To achieve this goal, a survey was conducted on a sample of 496 respondents living in the metropolitan area of Milan (North Italy). The declared preferences of the individuals for 12 strawberry characteristics, divided into intrinsic, extrinsic, and credence attributes, were first measured. A Latent Class Analysis (LCA) was then performed to identify different clusters of consumers according to the individuals’ preferences. Subsequently, the heterogeneity of the clusters was tested, using the Chi-square test, and sociodemographic characteristics and purchasing habits were considered. The results suggest that the most important attribute in the choice of strawberries was appearance, highlighting the importance of preserving it throughout the supply chain, followed by one of the increasingly important aspects of diets, which is health benefits. The attribute considered the least important was the brand. This study demonstrates, from a holistic point of view, that sociodemographic characteristics, food habits, and perceptions of different strawberry attributes influence consumers’ preferences and behaviors. Practical implications suggest a new prospective for communication marketing strategies for producers, creating a better brand identity and highlighting in their marketing all of the aspects that consumers would like to know about the fruits they choose as quality certifications.

## 1. Introduction

Soft fruits, which are becoming increasingly important in Italian diets, constitute a significantly segmented market that is influenced by various preferences and purchasing behaviors of the consumers [1,2].

Multiple factors are involved in determining purchasing choices in agri-food sectors, and they show a high level of difference depending on the characteristics of fruits and between different kinds of food products. Intrinsic factors such as physical characteristics, origin, variety, and nutritional attributes, as well as extrinsic factors not directly related to the quality of a fruit, such as brand and price, are considered important attributes of fruit during the purchase decision-making process. Often, certain intangible beliefs associated with the purchasing experience play an important role, like certifications and the perception of sustainability [3,4,5].

Based on previous research, different fruit consumption orientations have emerged: some consumers show greater interest in the health and nutritional aspects [6], taking care of their body, while others pay attention to the environmental impacts and seasonality of fruit [7], following the natural flow of nature. Finally, others attribute greater value to the aesthetic appearance and taste of the fruit.

Strawberries, despite being botanically classified as a false fruit, are one of the most consumed small fruits in Italy that convince Italian consumers. According to the data collected from Istituto di Servizi per il Mercato Agricolo Alimentare [8], a public economic research organization, Italian consumers’ spending on strawberry purchases increased in 2022 compared with both 2021 (+4.1%) and 2019 (+23.0%). From 2020 to 2021, volume demand increased by 4%, and in value, by 9%. This increase is due to the many production innovations, genetic enhancements, and technological improvements that have allowed this supply chain to penetrate markets even during winter. In Italy, four regions are known for their strawberry cultivations: Basilicata and Campania (which together account for 50% of the national surface area), Emilia-Romagna, and Veneto [9]. The strawberry farming sector in fact plays a strategic role in Italy. Despite their botanical categorization, strawberries are commercially classified as berries and are generally displayed in the same physical location as other berry fruits when sold.

Several international studies on strawberry consumption, preferences, and purchasing decisions were found in the literature. The attributes that drive the consumption of strawberries are generally sensory properties such as appearance, aroma, and flavor. This is why technological improvement improvements have been studied and applied in strawberry production to increase sweetness, enhanced by specific volatile compounds [1,10,11,12]. There are studies [13,14] that have explained the importance of smell and aroma during and after consumption, as well as how these factors could play an important role in a person’s choice of strawberries. Thanks to their intense sweet taste and versatility in culinary preparations as an ingredient of sweet or savory recipes, strawberries are becoming increasingly popular among consumers. Visual quality is one of the determining factors for consumers during the purchase and before the consumption of strawberries, and it encompasses various features, such as color, size, and freshness [15,16]. From chemical and nutritional research, strawberries show a unique blend of antioxidant and anti-inflammatory compounds, positioning them as potential allies in promoting cardiovascular health [17].

The innovations related to strawberries are not only related to genetic enhancements and postharvest techniques but also to distribution systems and packaging in terms of sustainable materials and size. This includes introducing new packaging methods aimed at preserving the fruit [18,19].

In recent years, alongside these traditional characteristics, new selection criteria have emerged that reflect a shift in consumers’ preferences. Among these, the search for locally sourced products is increasingly becoming prevalent. Several studies have shown that consumers associate the local origin of strawberries with better aesthetic attributes (such as color and freshness), better nutritional properties (which lead to enhanced health benefits), and the environmental sustainability of the fruit [15,20,21,22]. Sustainability, which is driven by the growing awareness of the environmental impacts of certain agricultural practices, has become one of the major trends in the food industry and has led to increased adoption of organic farming practices [23,24]. In Italy, there was a 34.0% increase in the volume of organic strawberry production in 2021, with 8.6% of the sold strawberries indicating organic certification, which is now a significant purchasing driver. Indeed, organic produce certification has become important for Italian consumers who are interested in high-quality fruits and vegetables [4]. A further aspect of importance, linked to sustainability, is seasonality, which has emerged as one of the most significant attributes for Italian consumers with regard to both strawberries and fruits in general [25,26]. The preference for seasonal strawberries is linked to their being associated with a greater degree of freshness and quality [4,27].

Understanding the consumption patterns of strawberries is essential to meet the evolving expectations and needs of consumers. This investigation is important to develop effective agricultural and marketing strategies and can contribute to a better matching of supply and demand.

Other studies have investigated strawberry consumers in the United States [21], applying class analysis, and some attributes have been found through conjoint analysis, or [2] consumers’ preferences have been explored by trained sensory panels, via Quantitative Descriptive Analysis (QDA), to establish their intrinsic attributes, but no studies have been conducted in Italy to profile the consumption patterns of Italians, considering their preferences together with their habits, their purchase consumption patterns, and their socio-demographic characteristics. In this context, this research was developed to obtain a holistic overview, that is, considering various attributes, including sensory aspects, as well as credence and extrinsic ones, to answer the following research questions: (1) What is the degree of importance that consumers attach to the different attributes that describe strawberries? (2) Are there different strawberry choice patterns that are heterogeneous from a sociodemographic and purchase preference point of view? (3) What role does certification play, compared with the other attributes, during the decision-making process?

The purchasing drivers of strawberries for Italian consumers have been identified and analyzed in this research to answer these questions. The Best–Worst Scaling (BWS) methodology was considered to assess the declared preferences expressed by consumers for a set of fruit attributes. This methodological approach allowed a high number of attributes, that is, sensory, credence, and extrinsic attributes, to be considered in the same experimental design to describe strawberries. In addition, a Latent Class Analysis (LCA) was applied to identify different preference profiles and to combine them with the individuals’ sociodemographic characteristics and purchasing habits.

## 2. Materials and Methods

### 2.1. Data Collection

Data were collected using a paper questionnaire in Italian language submitted randomly and face-to-face to individuals intercepted between July 2021 and October 2021 outside farmers’ markets and large-scale retail (LRd) stores distributed throughout the metropolitan area of Milan, Italy. The only inclusion criteria were related to the age of participants, who had to be over 18 years old. However, prior to participation, respondents were presented with an introductory document outlining the purpose of the research, the structure of the questionnaire, and assurances of anonymity and noncommercial use of the collected data. Additionally, participants were informed of their right to withdraw from the questionnaire at any time. In the end, we obtained 496 respondents participation. The research adhered to the principles of the Declaration of Helsinki, and the only requirement for participating in the survey was that the participants had to be at least 18 years old. About 8–10 min were required to fill in the questionnaire.

The questionnaire (Appendix A) was structured in three sections: (a) sociodemographic features, (b) strawberry consumption habits, and (c) consumers’ preferences assessment. The first section (sociodemographic) included several items related to the sociodemographic characteristics of the respondents, such as gender (female, male, or no answer), age (18–25, 26–35, 36–45, 46–55, 56–65, over 65), zip code of residence, household composition (number of components) and number of school-aged children, educational level (primary school, lower secondary school, upper secondary school, degree/postgraduate qualification), employment situation (student, employed, self-employed, retired, seeking employment, housemaker), and annual income range (<EUR 25,000, EUR 25,000–40,000, EUR 40,000–60,000, >EUR 60,000, or no answer). The second section investigated strawberry purchasing and consumption habits through three closed-ended questions: one question was related to the type of purchase retailers (greengrocer, open-air market, organic shop, directly from the producer, LRd stores, namely super/hypermarket or discount, grocery store, ethical purchasing groups), one was related to the seasonality of purchasing and consumption (spring, summer, autumn, winter, or over the entire year), and the last question was aimed at investigating the frequency of purchase (2–3 times a week, once/twice a week, less than once a week, once a month, less than once a month).

Finally, the last section was dedicated to the assessment of the consumers’ declared preferences pertaining to 12 strawberry attributes that were included in the BWS experimental design. The selected fruit characteristics were derived from in-depth literature research on consumers’ preferences (Table 1).

### 2.2. Best–Worst Scaling Methodology

The Best–Worst scaling methodology is a procedural approach that is used to gather people’s declared preferences related to a set of preselected attributes that characterize a product [4], in our case strawberries. This BWS experimental design was developed using Sawtooth MaxDiff Designer Software (SSI-version 8.4.6, Sawtooth Software, Orem, UT, USA).

The experimental design used in this research had already been used in other consumer behavior studies [55,56,57]. In accordance with the standard design commonly used in BWS surveys, given a set of n (n = 12) attributes of strawberries, r choice sets are provided (r = 9), and each set contains 4 attributes (t = 4) (constant condition n > t) that are differently combined, and each item appears 3 times in the questionnaire (s = 3), according to a balanced incomplete block scheme (Table 2) [58,59,60]. The respondents had to choose two attributes: the most important attributes (BEST) and the least important ones (WORST). To obtain a realistic average possible of attributes, the Sawtooth Software created 4 versions of the questionnaire: the first two sections were the same for all versions, while the last section (BWS) presented different combinational levels of attributes for each version. In this way, it was possible to obtain a high combination level of attributes, minimizing error. Participants were encouraged to consider a trade-off to simulate real-life decision making. By repeatedly asking the consumers to identify the most influential (best) and least influential (worst) attributes during product selection and purchase, the software calculated a mean preference index for each chosen item, using a probabilistic hierarchical Bayesian analysis approach. The time during which an item was designated as best or worst reflected the strength of preference for that particular attribute [61].

### 2.3. Statistical Analysis

The statistical analysis was organized in three main stages.

The data obtained for each attribute, which described the consumers’ strawberry preferences in terms of a couple of best–worst scores, were utilized to obtain the average raw score (ARS). This involved deriving the difference between the instances when each attribute was chosen as the worst and when it was selected as the best, and subsequently dividing it by the sample size and the number of times each attribute appeared in the questionnaire (which was 3 in our experimental design). In the second stage of the data elaboration, the objective was to group together consumers with similar preferences for strawberries by calculating the likelihood of belonging to each cluster together with their corresponding class-specific preference weights. This division of the entire sample into homogeneous classes (Clusters) was enabled by Latent Class Analysis (LCA) [62]. LCA does not allow the number and size of clusters to be known beforehand [63]. The software returns a number of possible subdivision combinations, but the best number of clusters is chosen, in agreement with Massaglia et al. [56], by looking for the lower value of the Bayesian Information Criterion (BIC) and the associated Log-Likelihood (LL) according to [64]. A segmentation with 5 clusters was chosen as the best approach. Each segment was characterized by precise sociodemographic and behavioral characteristics pertaining to strawberry purchasing. ANOVA and Turkey’s post hoc multiple comparison tests were conducted with SPSS 28.0 software and were used to assess heterogeneity over the five clusters, in terms of the declared strawberry attribute preferences, sociodemographic characteristics, and food habits.

Chi-square tests were used to define the independence of the different variables, and adding the elaborations of adjusted residuals made it possible to determine which cluster was statistically different for each variable. The last stage involved evaluating the correspondence between clusters and retailers through a corresponding analysis conducted with SPSS 28.0 software.

## 3. Results

Overall, 496 individuals declared they consumed and purchased strawberries. The whole sample was predominantly composed of women (69%) and small family units (42%, consisting of 2 individuals per family).

### 3.1. The Consumers’ Preferences for Strawberries

As reported in Table 3, the attribute of strawberries considered the most important by the consumers was “Appearance” (ARS = 3.333), followed by “Health benefits” (ARS = 2.847) and “Taste/aroma” (ARS = 2.224). The attribute considered the least relevant for strawberry selection was “Brand” (ARS = −3.109), followed by “Indication of origin (national/foreign)” (ARS = −2.112) and “Packaging” (ARS = −1.796). A negative ARS score indicates that the number of times the attributes were selected as “Best” was lower than the number of times they were chosen as “Worst”.

### 3.2. Latent Class Clustering Analysis

Five clusters were identified within the sample and were named according to the characteristics of strawberries that emerged as the most relevant during the purchasing decision moment (Table 4). The segmentation of the 5-cluster model met the selection criteria (Log-Likelihood = −8142.3566, BIC = 16821.39123). The ANOVA results show significant differences in preferences between the five clusters for all the strawberries (*p*-value < 0.001).

The characterization of the sample, in terms of sociodemographic characteristic variables and purchase and consumption variables, is reported in Table 5.

The results indicate significant differences among the clusters regarding the importance assigned to all the strawberry attributes examined in this study.

What emerges from our findings is the unbalanced distribution of the total sample across the clusters, with some clusters having a high percentage of the total size (Price Conscious and Health Conscious), and others having a relatively small percentage (Local Supporters, Nature Enthusiasts, Organic Excellence Supporters).

“Health Conscious” (33.5% of the sample) includes participants who expressed a significant interest in the aesthetic and sensory aspects of the fruit, as well as the potential health benefits associated with its consumption. These cluster characteristics were reflected in the purchasing and consumption habits of the interviewees. Indeed, most of them (45.0%) indicated they bought strawberries at least three times a week and preferred open-air markets and organic shops as their places of purchase. These behaviors reflect the economic profile of the cluster, with the majority reporting a medium-to-high annual income. A significant percentage of the interviewees (79.0%) were aged 46 or more, and they generally did not have school-aged children. The portion of male respondents (37.0%) in this cluster was higher than in the other four groups.

“Price Conscious”, the most numerous cluster (35.0%), was composed of respondents who paid greater attention to the price and to offers than the rest of the sample. This segment consisted of individuals with a medium-low income and below-average education levels. This consumer group preferred to purchase in LRd stores and open-air markets.

Members of the “Nature Enthusiasts” cluster were attracted to the visual aspect of strawberries because they associated it with the naturalness of the product. Quality certification was important for them in orienting their behavior. However, they gave little importance to the origin. Moreover, they were the most inclined to respect the seasonality of the products, and predominantly consumed strawberries in spring and summer (82.0% of the cluster), albeit less frequently than the overall sample.

The members of the “Organic Excellence Supporters” cluster attributed more importance to the organic certification of strawberries than the other clusters. They believed that the organic origin of the product and its appearance contributed to the superior organoleptic and nutritional quality of strawberries. These attribute preferences may be linked to the high level of education and income of the cluster members. Furthermore, their attention to product quality reflects their search for certified and safe strawberries for their children. In fact, this cluster had the highest percentage of individuals with school-aged children (32.0%). They preferred buying strawberries from organic shops, but not so much from open-air markets because it was less likely they would find certified products there.

“Local Supporters”, the smallest cluster (6.0%), was composed of the highest percentage of young individuals (29.0% under 46 years of age), who paid more attention to sustainability-related features than the rest of the sample. The attribute they considered the most important was the indication of the origin of strawberry productions, especially those of local origin. These traits indicated a greater awareness of product certifications, so this group was the only one that assigned significant importance to brands. The significance of brands is linked to the consumers belief that quality is ensured when they choose a local brand. Additionally, they are more inclined to purchase from LRd, presumably because that is where they found their preferred brands, but also from greengrocers and open-air markets, where they were sure of finding local strawberry producers.

## 4. Discussion

In this study, the strawberry preferences of consumers have been studied through the Best–Worst Scaling methodology to identify the most relevant factors in strawberry purchasing behavior. In addition, different purchasing and preference profiles have been identified and investigated.

Unlike our results, the research of Bhat et al. [14] showed a greater interest of consumers in the taste of strawberries. However, our investigation suggests that the appearance of strawberries is the most important attribute for consumers during the decision-making process, and taste was only ranked third, being preceded by health benefits. This difference, in terms of ranking, is not so relevant, as taste and appearance demonstrate a close correlation: the appearance of strawberries, including such factors as texture, hydration level, and color, plays an important role in influencing consumers’ perception of flavor, taste, and aroma. This ranking is justified if we consider that the first sense that comes into play when a product is introduced to a consumer is sight, thus color and defects are immediately perceived at the moment of purchase [65]. The integrity of the peel is closely associated with the ripeness stage and with the absence of biotic and abiotic defects [11]. It was observed that a poor appearance means poor flavor characteristics and taste [66]. Indeed, a good appearance of strawberry peels and leaves contributes to a higher perception of freshness [16]. This is particularly true when not only strawberries are considered but also other small fruits, such as blueberries, raspberries, grapes, cherry tomatoes, and Actinidia arguta. Indeed, no peeling is necessary for the berry category before eating and, thus, any imperfection present on the rather thin cuticle layer is not removed [67]. It seems that consumers obtain different information about taste and aroma from the appearance of strawberries and consider these attributes crucial during the purchasing process. Several studies, such as [68,69], have highlighted the high nutritional value of strawberries. Our own research findings confirm that the health benefits of strawberries significantly influence consumption choices. Moreover, in this case, the appearance appears to play a role in presenting the nutritional qualities of strawberries, because their color is due to polyphenolics, such as anthocyanins, which are present in fruits and possibly related to health benefits. Consumers are becoming more and more aware of the nutraceutical aspects of fruits and are influenced by such considerations during the purchasing phase [70,71]. Additionally, a research investigation into the purchasing preferences of strawberries among German consumers by Bhat et al. [14] revealed a rising awareness of the health benefits associated with the consumption of strawberries. This awareness is aligned with a wider trend of increased attention to the nutritional qualities of regularly consumed food [69,72,73].

According to the literature [21], an indication of origin and branding could provide a point of differentiation for local producers. However, our findings suggest that these attributes are not considered to be so important. Taylor k. Ruth’s investigation [74] into the importance of branding berries in the United States highlighted that local brands are preferred because they indicate the production origin, and they thus indicate a positive value or meaning. However, there may be a lack of well-known strawberry brands in Italy that offer distinct qualities, thus making it challenging for consumers to develop a specific preference.

Although the identified clusters do not show an equal distribution over the sample, the results seem to offer an accurate representation of Italians’ consumption and purchasing profiles of strawberries. Five clusters were identified, but a surface-level examination also allowed two styles of consumers to be identified: health-conscious and price-conscious ones. These categories represent the classic consumption styles, which are quite widespread among consumers (together, they make up almost 70.0% of our sample). Meanwhile, Nature Enthusiasts, Organic Excellence Supporters, and Local Supporters represent modern consumption styles, which are linked to environmental or personal health concerns. In spite of being smaller in size, they present well-defined and specific decision-making profiles that clearly stand out from the two main groups. Over the last decade, other studies that have investigated fruit and vegetable consumption styles [4,75] have revealed a growing interest in the environment and health benefits of food.

Considering the consumption habits and sociodemographic characteristics as variables, significant characteristics of the grouping emerged and appear to be in accordance with the consumption profiles. The largest group was composed of price-conscious members, that is, people who are interested in low prices and offers. As Aoki et al. [10] found in their research, consumers preferred to buy cheaper white strawberries, and the same approach could be considered valid for traditional strawberries. It has emerged, from our findings, that the price-conscious interviewees preferred to purchase strawberries in open-air markets and in greengrocers, probably because they could find low-cost strawberries in these places. However, some studies [76,77,78] that have investigated the purchase of fruits and vegetables in open-air markets have revealed different results from ours: many consumers purchased food products from open-air markets in order to contribute to the local economy [79], but one problem they encountered was excessive prices. This difference in outcomes between the present study and those mentioned might be attributed to how an outdoor market is structured. In an open-air market, it is possible to find both high- and low-priced strawberries. The lower transportation and marketing costs of open-air markets than traditional channels lead to more competitive price options; however, high-quality products, even if they are local, might still be expensive.

The Local Supporters, who preferred purchasing local brand strawberries, predominantly chose greengrocers, where it is common to find fruits and vegetables produced and sold in the same place/geographical zone as that of purchase [77]. It is not surprising that the members of this group were younger. Evidence that has emerged from the literature suggests that the younger generations are generally considered to be more progressive in promoting sustainability and adopting a proenvironmental lifestyle [80].

The Health Conscious found strawberries very attractive due to their antioxidant and anti-inflammatory properties [81], which offer significant health benefits. The heightened interest in the aesthetic attribute of strawberries, as already discussed, which is more pronounced for Health Conscious, has given us additional information. In this case, members of this group looked at the appearance of the skin and integrity of the fruit to gather health-related information. Unlike other fruits, the skin of strawberries can be easily damaged, thus leading to the development of mold and diseases and consequently to variations in the texture that could translate into a poor-quality fruit with an unhealthy appearance [82]. Consequently, they preferred to purchase strawberries from organic shops or open-air markets, even though they were costlier, as they can offer strawberries that adhere to certain food standards [83].

Furthermore, the Organic Excellence Supporters usually bought from organic shops but for a different reason: they believed the possibility of finding organic certifications and quality certifications ensured they were purchasing high-quality strawberries. Certifications and logos ensured consumers that the product adhered to specific standards and that it was safe [84]. According to Gundala et al. [85], people with higher levels of education are more inclined to purchase organic food because they have a better understanding of its significance and a heightened awareness of certification logos. The members of this group had more households with children; thus, they chose safe food options for their children. However, another research has found that the consumption of organic food decreases when children become adolescents [86].

The importance that Nature Enthusiasts gave to the seasonality of strawberries suggests a high level of awareness of the natural growth cycle; they preferred to consume strawberries in harmony with the rhythms of nature and considered the preservation of the environment. According to Wallnoefer et al. [87], such participants recognized that their choice was considered to be of an ecologic type, which favored biodiversity, supported the local economy and farmers, and promoted dietary diversity. Although the data in our study were collected in the metropolitan area of Milan, which is an urban setting, according to Spence et al. [88], humans are still influenced by climatic conditions, and thus, seasonality remains evident in our contemporary eating behaviors.

## 5. Conclusions

This study aims to explore the declared preferences of consumers of strawberries considering different characteristics and attributes from a holistic point of view. The main outcome has been highlighted. Appearance was identified as the main driver of purchasing strawberries, followed by health benefits and taste/aroma. This research identified five different clusters defined by the consumers’ purchasing habits and sociodemographic characteristics and preferences in strawberries: nearly 70% of respondents fell into health-conscious and price-conscious clusters, and the highest percentage of young people who considered sustainability and environment-friendly characteristics important was found in the Local Supporters cluster. The Nature Enthusiasts, Organic Excellence Supporters, and Local Supporters represent clusters that had a relatively small proportion of members compared with the overall sample. This indicates that although the individuals shared similar purchasing and consumption behaviors, and certain sociodemographic characteristics were few in number, they still exhibited distinct and well-defined consumption patterns.

The great importance of appearance for consumers indicates the need to promote technological development in breeding, elevate attention to logistics in the product distribution stages, and enhance postharvest care. By integrating new conservation techniques into the distribution system, it will be possible to ensure a better appearance of strawberries, preserving not only the aesthetic appearance but reducing the occurrence of fruit diseases. This could be of interest to the market because it could help to reduce waste and encourage producers to orient strawberry offers toward strawberry preferences; consequently, businesses can enhance competitiveness and improve the communication strategies in terms of all the attributes that were under consideration by consumers. Moreover, targeted communication, based on the consumers’ priorities, could be useful for producers to satisfy consumers by anticipating their needs and expectations.

The Best–Worst Scaling methodology applied in this study yielded valuable and practical results: the experimental design allowed us to maximize the information collected while effectively reducing participants’ cognitive efforts during data collection. Although this study has revealed the significant attributes influencing consumers’ preferences for strawberries, limitations exist that are primarily related to the geographical boundaries chosen for the sample population because the results could be conditioned by narrowing cultural and climatic factors. However, future studies could address this by employing various sampling methods to broaden the scale. For instance, future studies could delve deeper into consumer evaluations, particularly regarding credence and beliefs. Moreover, future studies could consider different geographic areas in terms of individuals residing in both small and large urban areas in various Italian regions.

## Figures and Tables

**Table 1 foods-13-01474-t001:** Strawberry attributes analyzed by the Best–Worst scaling methodology.

Attribute Category	Strawberry Attributes	Description	References
Extrinsic attributes	Brand	The brand enables consumers to identify a product and even prefer it over others.	[28,29]
	Indication of origin (national/foreign)	The origin of the product and the producer can be traced back to the concept of local food.	[30,31]
	Price	The price can be a decisive factor during the decision-making process. It is linked to a consumer’s willingness to pay for a product.	[32,33]
	Packaging	Packaging, especially such factors as dimension and material, can be important for consumers, for example, it may be related to sustainability issues.	[34,35,36,37]
	Offers	Offers can be an important factor, such as a promotion, especially for products with a medium-to-high price.	[38,39]
Intrinsic attributes	Appearance	The aspects of strawberries, especially in terms of integrity of the skin, texture, and shape. In the vegetable sector, the appearance of the product can be a sign of freshness.	[2,16]
	Taste/Aroma	Flavor has the ability to evoke emotions and influence purchases.	[40,41]
Credence attributes	Health benefits	The health characteristics and the benefits derived from their consumption are considered important; they are often considered a “superfood”.	[42,43,44,45]
	Local Origin	Local production is synonymous with a short supply chain and thus sustainability for the consumer.	[46,47,48]
	Organic certification	When a product is certified as organic, it means that it adheres to certain production standards. This certification can influence consumers’ choices.	[49,50]
	Quality certification	Quality certification conveys confidence and indicates specific quality standards.	[51,52]
	Seasonality	Seasonality is crucial for fresh products and indicates the importance of consuming fruit in season.	[53,54]

**Table 2 foods-13-01474-t002:** Example of a Best–Worst Scaling question: the respondent has to choose the best and the worst attributes.

Least Important (Only One Choice)	Strawberry Attributes	Most Important (Only One Choice)
()	Brand	()
()	Price	()
()	Local Origin	()
()	Taste/aroma	()

**Table 3 foods-13-01474-t003:** The times an attribute was selected as the best, the times an attribute was selected as the worst, and the average raw score (ARS) of each strawberry attribute.

Rank	Strawberry Attributes	Time Selected Best	Time Selected Worst	B-W Score	BW Average Raw Score (ARS)
1	Appearance	1009.0	35.0	974.0	3.333
2	Health Benefits	885.0	48.0	837.0	2.847
3	Taste/Aroma	741.0	102.0	639.0	2.224
4	Quality certification	408.0	329.0	79.0	0.368
5	Price	383.0	275.0	108.0	0.317
6	Offers	328.0	332.0	−4.0	0.047
7	Seasonality	167.0	267.0	−100.0	−0.257
8	Organic Certification	182.0	381.0	−199.0	−0.812
9	Local Origin	129.0	434.0	−305.0	−1.048
10	Packaging	107.0	604.0	−497.0	−1.796
11	Indication of Origin (national/foreign)	91.0	696.0	−605.0	−2.112
12	Brand	34.0	961.0	−927.0	−3.109

**Table 4 foods-13-01474-t004:** Analysis of variance (ANOVA): strawberry attribute preferences in the five clusters.

Cluster	Health Conscious	Price Conscious	Nature Enthusiasts	Organic Excellence Supporters	Local Supporters	F	*p*-Value
*Cluster dimension*	33.50%	35.00%	15.40%	10.20%	5.90%		
**Attributes**	** *Average BW raw scores* **						
Brand	1.055 ^b^	0.507 ^a^	0.997 ^b^	0.544 ^b^	8.451 ^c^	81.614	***
Health Benefits	22.543 ^c^	18.924 ^c^	15.097 ^b^	13.931 ^b^	9.684 ^a^	81.061	***
Appearance	22.707 ^b^	19.719 ^b^	21.363 ^b^	19.951 ^b^	9.889 ^a^	66.441	***
Indication of Origin (national/foreign)	3.319 ^b^	1.162 ^a^	1.229 ^a^	0.534 ^a^	7.849 ^c^	45.195	***
Local Origin	3.686 ^a,b^	2.693 ^a^	4.469 ^b^	2.899 ^b^	9.114 ^c^	23.984	***
Organic certification	4.158 ^b^	2.134 ^a^	2.641 ^b^	16.993 ^d^	12.561 ^c^	113.459	***
Price	5.116 ^c^	15.363 ^d^	5.400 ^c^	1.353 ^a^	4.056 ^b^	190.987	***
Quality Certification	5.147 ^a^	3.727 ^a^	21.329 ^c^	15.928 ^c^	12.746 ^b^	130.200	***
Packaging	1.760 ^b^	1.075 ^a^	3.801 ^c^	9.249 ^d^	7.756 ^d^	100.699	***
Taste/Aroma	21.821 ^e^	15.199 ^d^	9.198 ^b^	14.056 ^c^	6.064 ^a^	100.742	***
Offers	3.661 ^c^	14.970 ^d^	4.832 ^c^	1.268 ^a^	3.264 ^b^	207.817	***
Seasonality	5.029 ^a,b^	4.528 ^a,b^	9.643 ^c^	3.295 ^a^	8.567 ^b,c^	10.215	***

^a,b,c,d^ The preference averages (rescaled scores) within a row with the same letters are statistically different (_ = 0.05, Tukey’s post-hoc test). The *p*-value refers to the statistical significance level: *** < 0.001.

**Table 5 foods-13-01474-t005:** Differences in the sociodemographic characteristics and purchase and consumption habits between the five clusters (X-squared test). The bold-highlighted cluster indicates significant differences from others for each considered variable.

Cluster		Health Conscious	Price Conscious	Nature Enthusiasts	Organic Excellence Supporters	Local Supporters	X-Squared	*p*-Value
*Socio-demographic variables*								
Gender	Female	67.2	70.6	73.3	76.0	78.6	6.361	n.s
	Male	37.3	29.4	26.7	24.0	21.4		
Age	18–25	0.0	1.7	0.0	2.0	**3.6**	42.210	**
	26–35	6.0	2.8	12.0	8.0	7.1		
	36–45	15.1	13.0	18.7	18.0	17.9		
	46–55	30.7	31.1	30.7	**50.0**	46.4		
	56–65	27.1	24.9	29.3	20.0	10.7		
	Over 65	21.1	**26.6**	9.3	2.0	14.3		
Household composition	1 component	16.3	16.9	9.3	8.0	7.1	16.554	n.s
	2 components	40.4	41.2	50.7	34.0	42.9		
	3 components	24.7	28.8	24.0	30.0	35.7		
	4 components	15.7	10.7	14.7	26.0	10.7		
	5 components	3.0	2.3	1.3	2.0	3.6		
School-aged children	No	74.1	75.1	73.3	68.0	71.4	11.659	n.s
	Yes, 1	18.7	18.1	18.7	24.0	25.0		
	Yes, 2	6.0	3.4	8.0	8.0	0.0		
	Yes, more than 2	1.2	3.4	0.0	0.0	3.6		
Education	Primary school	4.2	**8.5**	5.3	0.0	0.0	36.383	***
	Lower secondary school	17.5	**26.0**	14.7	6.0	14.3		
	Upper secondary school	57.8	52.5	50.7	54.0	53.6		
	Degree/Post-graduate qualification	20.5	13.0	29.3	**40.0**	32.1		
Employment situation	Student	0.0	2.3	2.7	2.0	**3.6**	68.475	***
	Employed	47.6	42.9	57.3	48.0	42.9		
	Self-employed	15.1	5.6	14.7	**34.0**	17.9		
	Retired	20.5	**28.2**	8.0	4.0	14.3		
	Seeking employment	1.8	6.2	6.7	6.0	**10.7**		
	Housemaker	14.5	14.1	8.0	6.0	10.7		
Income range (€)	<25,000	18.7	**30.5**	20.0	10.0	17.9	60.206	***
	25,000–40,000	49.4	57.1	40.0	36.0	39.3		
	40,000–60,000	27.7	10.7	36.0	44.0	39.3		
	>60,000	4.2	0.0	4.0	**6.0**	3.6		
	No answer	0.0	1.7	0.0	**4.0**	0.0		
*Purchase and consumption variables*								
Place of purchase of strawberries	Greengrocers	14.8	18.1	17.1	13.6	15.9	2.487	n.s
	Open-air market	35.5	38.1	32.2	**26.2**	34.8	15.060	**
	Organic shop	6.5	1.7	6.2	**13.6**	8.7	28.063	***
	Directly from the producer	0.3	0.3	1.4	1.9	**2.9**	10.009	*
	LRd stores (super/hypermarket, discount)	36.4	36.2	36.3	36.9	30.9	0.630	n.s
	Grocery stores	6.5	5.4	6.8	7.8	7.2	1.809	n.s
	Ethical purchasing groups	0.0	0.3	0.0	0.0	0.0	1.806	n.s
Consumption season of fresh strawberries	Spring	36.0	42.3	47.4	37.0	29.9	5.124	n.s
	Summer	25.2	23.2	**31.6**	26.9	26.4	13.483	**
	Autumn	11.0	9.8	**4.5**	10.1	12.6	16.159	**
	Winter	16.9	15.2	11.3	16.8	**18.4**	17.806	**
	All year	11.0	9.5	**5.3**	9.2	12.6	14.792	**
Number of strawberry purchases during the consumption period	2 times	10.8	**9.0**	21.3	16.0	21.4	23.608	*
	3 times	44.6	46.9	57.3	48.0	**32.1**	13.483	**
	4 times	33.7	31.6	**17.3**	32.0	39.3	16.159	**
	5 times	10.8	**12.4**	4.0	4.0	7.1	17.806	**

The *p*-value refers to the statistical significance level: *** < 0.001, ** < 0.01, * < 0.05, n.s.: not significant.

## Data Availability

The data presented in this study are available on request from the corresponding author. The data are not publicly available due to privacy restrictions.

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
