# Peer review of "Analyzing Strawberry Preferences: Best–Worst Scaling Methodology and Purchase Styles"

_foods, 2024, doi:10.3390/foods13101474_

Round 1
Reviewer 1 Report
Comments and Suggestions for Authors
I consider that the manuscript presents interesting information on Italian consumers' preferences towards strawberries using the best-worst scaling methodology. Some of my comments are the following:
The introduction should be better structured, information on the methodology and results are presented in this section which should be written in the corresponding sections.
The methodology should detail how the BWS was applied with respect to the sets that consumers had to answer, how many sets were designed for each participant?
In Table 5, it is suggested to incorporate the Chi-Square Per Cell to identify the source of variation of the cells in the variables that were statistically significant. As well as highlighting this information in the description of the groups identified.
Author Response
Reviewer 1
Rev: I consider that the manuscript presents interesting information on Italian consumers' preferences towards strawberries using the best-worst scaling methodology. Some of my comments are the following:
The introduction should be better structured, information on the methodology and results are presented in this section which should be written in the corresponding sections.
Author: Thank you for understanding the importance of the topic and for the constructive suggestions. We added clearer information: we improved the introduction following your comment and reorganizing the content.
The methodology should detail how the BWS was applied with respect to the sets that consumers had to answer, how many sets were designed for each participant?
Authors: Thank you for the question, we agree with you and we implemented the method among this topic in line 199-204 (track change on).
In Table 5, it is suggested to incorporate the Chi-Square Per Cell to identify the source of variation of the cells in the variables that were statistically significant. As well as highlighting this information in the description of the groups identified.
Authors: Thank you for the suggestion. According to adjusted residuals in crosstab, cluster significant different were identified for each variable. Additional information were added in line 241-243 (track change on).
Reviewer 2 Report
Comments and Suggestions for Authors
This manuscript presents important and relevant research examining fresh strawberry consumption preferences and purchasing patterns among consumers in a metropolitan area in northern Italy. For the purpose of this research, the Best-Worst Scaling (BWS) method was used to assess stated preferences for fruit attributes. Alternatively, latent class analysis (LCA) was used to identify preference profiles and combine them with the sociodemographic characteristics and purchasing habits of individuals. Their research also investigates the role of certifi cation relative to other attributes in the decision-making process.
Although the title describes the research in detail, it is too long. It would be better to come up with a smarter, more attention-grabbing, shorter title that covers the research. The keywords should be more specific and take the opportunity to include more keywords.
In the Abstarct section, more emphasis could be placed on specific research findings.
In the introduction section, it would be useful to provide recent, concrete consumption data, showing in more detail the quantities and proportions of fruit consumed by Italian and European consumers, especially for strawberries. There are several international studies on strawberry consumption, preferences and purchasing decisions, which could be better integrated in the introductory section, in particular those on chemicals, genetic modification, packaging or different uses of strawberries.
The objectives are appropriate. This paragraph should be moved to another part of the introduction: The survey was conducted in Italy, and in particular in the metropolitan city of Milan which, in 2021, recorded the highest percentage of strawberry purchases in the Northwest of Italy [10].
The Materials and methods section is generally acceptable, but I have a few comments on it. Data were collected using a paper-based questionnaire administered randomly and face-to-face to individuals in front of farmers' markets and large retail (LRd) stores in the metropolitan area of Milan. In the manuscript, it is claimed that this was chosen to make the answers more honest. It would be worthwhile to describe how the answers are more honest (on-line survey?), and to justify why consumers were randomly surveyed and how this was achieved.
You may wish to include the questionnaire as supplementary material.
Best-Worst scaling methodology and Latent Class Analysis (LCA) are proven methodological choices for this type of research. The three parts of the statistical section are described in sufficient detail. Given the justification for choosing the Bayesian Information Criterion to determine the number of clusters.
Table 3 illustrates that the results are presented in descending order based on the BW Aver-age Raw Score (ARS). The naming and characterization of the 5 clusters and the presentation of significant differences between clusters are appropriate.
The discussion section is one of the most valuable, well explained and written parts of the manuscript. Previous international research is compared in detail with our own results.
The Conclusion section could usefully be more concise and should integrate methodological lessons learned in addition to limitations and practical results. Future research directions should be better emphasised in this section.
Author Response
Reviewer 2
This manuscript presents important and relevant research examining fresh strawberry consumption preferences and purchasing patterns among consumers in a metropolitan area in northern Italy. For the purpose of this research, the Best-Worst Scaling (BWS) method was used to assess stated preferences for fruit attributes. Alternatively, latent class analysis (LCA) was used to identify preference profiles and combine them with the sociodemographic characteristics and purchasing habits of individuals. Their research also investigates the role of certification relative to other attributes in the decision-making process.
Although the title describes the research in detail, it is too long. It would be better to come up with a smarter, more attention-grabbing, shorter title that covers the research.
Authors: Thank you, we changed it.
The keywords should be more specific and take the opportunity to include more keywords.
Authors: Thank you. We added choice experiment; fruit aesthetics; health benefits; taste.
In the Abstract section, more emphasis could be placed on specific research findings.
Authors: Thank you, we improved it.
In the introduction section, it would be useful to provide recent, concrete consumption data, showing in more detail the quantities and proportions of fruit consumed by Italian and European consumers, especially for strawberries. There are several international studies on strawberry consumption, preferences and purchasing decisions, which could be better integrated in the introductory section, in particular those on chemicals, genetic modification, packaging or different uses of strawberries.
Authors: Thank you for your suggestion. Additional consumption data, particularly focusing on Italy, where the study was conducted, have been incorporated (line 58-66). Moreover, information on innovations in packaging and technological advancements has been included (line 93-95 and 101-104).
The objectives are appropriate. This paragraph should be moved to another part of the introduction: The survey was conducted in Italy, and in particular in the metropolitan city of Milan which, in 2021, recorded the highest percentage of strawberry purchases in the Northwest of Italy [10].
Authors: Thank you, we removed this paragraph because it was already explained in line 154-164.
The Materials and methods section is generally acceptable, but I have a few comments on it. Data were collected using a paper-based questionnaire administered randomly and face-to-face to individuals in front of farmers' markets and large retail (LRd) stores in the metropolitan area of Milan. In the manuscript, it is claimed that this was chosen to make the answers more honest. It would be worthwhile to describe how the answers are more honest (on-line survey?), and to justify why consumers were randomly surveyed and how this was achieved.
You may wish to include the questionnaire as supplementary material.
Authors: Thank you very much for pointing out this critical point. We agree with your note: such information creates confusion. We start therefore preferred to remove it. Data collection was randomized, as selections outside of supermarkets and retail settings were entirely random, adhering solely to the inclusion criteria. Because was not clear, we specify inclusion criteria.
We included the survey in the supplementary material.
Best-Worst scaling methodology and Latent Class Analysis (LCA) are proven methodological choices for this type of research. The three parts of the statistical section are described in sufficient detail. Given the justification for choosing the Bayesian Information Criterion to determine the number of clusters.
Authors: Thank you. By default, the Sawtooth software generates 4 segmentations, each containing the division of the sample into 2 to 5 clusters respectively. To determine the most suitable segmentation for our case study, certain indicators were considered. We followed the study conducted by:
Dekhili, S., Sirieix, L., & Cohen, E. (2011). How consumers choose olive oil: The importance of origin cues. Food quality and preference, 22(8), 757-762.
For this study, the Bayesian Information Criterion was chosen due to its lower number. There was another combination of segmentation that was unwrap:
|
Groups |
Log-likelihood |
BIC |
|
2 |
-8555,60757 |
17320,44493 |
|
3 |
-8376,68257 |
17071,75830 |
|
4 |
-8249,44844 |
16926,45343 |
|
5 |
-8142,33566 |
16821,39123 |
This is why 5 clusters were reported in the paper.
Table 3 illustrates that the results are presented in descending order based on the BW Aver-age Raw Score (ARS). The naming and characterization of the 5 clusters and the presentation of significant differences between clusters are appropriate.
The discussion section is one of the most valuable, well explained and written parts of the manuscript. Previous international research is compared in detail with our own results.
Authors: Thank you.
The Conclusion section could usefully be more concise and should integrate methodological lessons learned in addition to limitations and practical results. Future research directions should be better emphasized in this section.
Authors: Thank you. Conclusion section was improved considering your interesting suggestion to emphasize on the imprecation and highlight also the strength of the method.